# Black Phosphorus Nano-Polarizer with High Extinction Ratio in Visible and Near-Infrared Regime

**DOI:** 10.3390/nano9020168

**Published:** 2019-01-29

**Authors:** Wanfu Shen, Chunguang Hu, Shuchun Huo, Zhaoyang Sun, Guofang Fan, Jing Liu, Lidong Sun, Xiaotang Hu

**Affiliations:** 1State Key Laboratory of Precision Measuring Technology and Instruments, Tianjin University, Weijin Road, Tianjin 300072, China; 2Nanchang Institute for Microtechnology of Tianjin University, Weijin Road, Tianjin 300072, China; 3Institute of Experimental Physics, Johannes Kepler University Linz, A-4040 Linz, Austria; 4Key Laboratory of All Optical Network and Advanced Telecommunication Network of Ministry of Education, Institute of Lightwave Technology, Beijing Jiaotong University, Beijing 100044, China

**Keywords:** Low-symmetrical 2D materials, black phosphorus, resonance cavity, scattering-matrix calculation, nano polarizer

## Abstract

We study computationally the design of a high extinction ratio nano polarizer based on black phosphorus (BP). A scattering-matrix calculation method is applied to compute the overall polarization extinction ratio along two orthogonal directions. The results reveal that, with a resonance cavity of SiO_2_, both BP/${\mathrm{SiO}}_{2}$/Si and *h*-BN/BP/${\mathrm{SiO}}_{2}$/Si configurations can build a linear polarizer with extinction ratio higher than 16 dB at a polarized wavelength in the range of 400 nm–900 nm. The polarization wavelength is tunable by adjusting the thickness of the BP layer while the thicknesses of the isotrocpic layers are in charge of extinction ratios. The additional top layer of *h*-BN was used to prevent BP degradation from oxidation and strengthens the practical applications of BP polarizer. The study shows that the BP/${\mathrm{SiO}}_{2}$/Si structure, with a silicon compatible and easy-to-realize method, is a valuable solution when designing polarization functional module in integrated photonics and optical communications circuits.

## 1. Introduction

In-plane anisotropic two-dimensional (2D) materials, such as black phosphorus (BP) [1,2,3,4], rhenium disulfide (${\mathrm{ReS}}_{2}$) and rhenium diselenide (${\mathrm{ReSe}}_{2}$) [5,6,7], have recently drawn significant attention for versatile photonic and optoelectronic applications. Their optical anisotropy offers a valuable opportunity to control and manipulate the polarization state of light, which is a benefit for achieving highly efficient, compact and integrated all-optical light-processing operations [8,9]. As the first studied anisotropic 2D materials, BP exhibits strong optical anisotropy and nonlinear optical response, making it a preferred 2D materials for optical saturable absorber, particularly for ultrafast pulse laser and switching devices [10,11,12]. Recently, the prototypes of BP-based light generators [11,12,13,14], modulators [15,16,17,18,19], sensors [20,21] and detectors [22,23,24] have been proposed, and their performances keep improving.

Interestingly, the numerical difference between refractive indices of two crystalline axes of BP flake (e.g., Δn=0.19 at a wavelength of 600 nm) is even bigger than the calcite (∼0.172), which is rare in the family of optical materials and makes BP suitable for building optical polarization device. In contrast to an abundant of investigations on the optical modulators based on graphene and TMDs [25,26,27], BP optical modulators are less investigated despite several limited prototypes. For example, BP-coated microfiber was demonstrated by J. Zheng et al. [18], which is used as optical Kerr switcher and four-wave-mixing-based wavelength converter. F. Zhou et al. studied BP polarizer based on BP/${\mathrm{SiO}}_{2}$ metamaterials waveguide in theory [28]. Most recently, we proposed a prototype of BP polarizer based on the Fabry-Perot cavity of ${\mathrm{SiO}}_{2}$, which is simpler and easier for integration compared with other techniques, like plasmonics and metasurface [19,25,29,30]. The resonance cavity is an effective approach to increase the interaction between the light and the anisotropic materials. However, how to design a nano polarizer with high extinction ratio (ER) is still not clear in theory.

In this paper, we analyzed the design for high-performance BP polarizer in Fabry-Perot configuration. We started from three-phase system (air-BP-substrate) and derived the conditions for the refractive index of the substrate for high polarization effect. Interestingly, the suspended BP thin film in the air with thickness of 1st order destructive interference (∼70 nm) exhibits strong polarization effect with ER around 10 dB. In contrast, when depositing BP flake directly on a silicon or quartz substrate, it shows a weak polarization effect because of the mismatches of refractive indices of the two layers. When Fabry-Perot cavity is added, such as four-phase-model of air/BP/${\mathrm{SiO}}_{2}$/Si and five-phase model of air/*h*-BN/BP/${\mathrm{SiO}}_{2}$/Si, a huge polarization effect with extinction ratio higher than 16 dB is achievable easily. The polarization wavelength can be continuously selected in a broad range from 400 nm to 900 nm. Furthermore, we explored dependences of polarization wavelength and extinction ratio on the thicknesses of BP and ${\mathrm{SiO}}_{2}$ layers, respectively. The calculation reveals that the thickness of the anisotropic layer (BP) is determinant for the polarization wavelength and the thickness of the isotropic layer (${\mathrm{SiO}}_{2}$) has has strong effect on the extinction ratio. The ∼50 nm ${\mathrm{SiO}}_{2}/\mathrm{Si}$ is the most suitable substrate on which the polarization wavelength shows mildest dependence on the thickness of BP layer. Besides, to overcome the oxidization of BP, we further explored the optical anisotropy $∆R/R_{AZ}$ of air/*h*-BN/BP/${\mathrm{SiO}}_{2}$/Si system as the thickness of *h*-BN at 6 nm and 30 nm, respectively, and concluded the optimized parameters for building a polarizer with ER higher than 16 dB.

## 2. Materials and Methods

Figure 1a schematically shows the working principle of a nanoscale polarizer based on an interference effect of an anisotropic film. The light passes through the anisotropic film with different phase difference at different orientations because divergent refraction indices. When the phase difference δ along one of the crystal axes equals 2mπ(m=1,2,3…), the reflectance along this direction ($R_{\left|\right|}\;or\;R_{⊥}$) is extremely suppressed. At the same time, the reflected light along the other crystal axis exists because the divergent complex refractive indices ($\tilde{n_{i}}$). Through an elaborate design, it can be used to block the reflectance along one axis and pass the reflectance along its orthogonal axis. This unique phenomenon provides excellent opportunity to regulate the polarization of the light with the new rising anisotropic 2D materials, such as BP, ReS2 and so on.

Figure 1b denotes the proposed nanoscale polarizer based on BP-cavity-substrate heterostructures. The anisotropic BP layer induces optical anisotropy. The transparent cavity enhances the light interactions through resonance effect. To quantify the polarization degree of the heterostructure, we define the optical anisotropy $∆R/R_{AZ}$ as:
(1)$$∆R/R_{AZ}=2\frac{R_{AC}-R_{ZZ}}{R_{AC}+R_{ZZ}}\equiv 2N,$$where $R_{AC}$ and $R_{ZZ}$ symbol optical reflectances along armchair (AC) and zigzag (ZZ) directions of BP, respectively. According to the definition, $∆R/R_{AZ}$ varies from −2 to +2 and goes to extreme value ±2 when one of the reflectances approaches zero.

In the computational analysis, we independently calculated reflectances along the two orthogonal crystal axes ($R_{AC}$ and $R_{ZZ}$) of the multilayer heterostructure and obtained the optical anisotropy value in Equation (Equation 1). Using scattering matrices method, the overall reflection and transmission coefficients of the stratified system are
(2)$$r_{p}=\frac{S_{21}^{p}}{S_{11}^{p}},\;t_{p}=\frac{1}{S_{11}^{p}},$$where S are the total scattering matrices of a nano polarizer and *p* indicates the AC or ZZ directions. The reflectance intensity ($R_{ZZ}$ or $R_{AC}$) was obtained through multiplying the reflection coefficients $r_{p}$ by its complex conjugate.

The total scattering matrices S along the principal axes were calculated by
(3)$$S^{p}=I_{01}^{p}\cdot L_{1}^{p}\cdot I_{12}^{p}\cdot L_{2}^{p}\ldots I_{(j-1)j}^{p}\cdot L_{j}^{p}\cdot I_{j(j+1)}^{p}\ldots I_{(m-1)m}^{p}\cdot L_{m}^{p}\cdot I_{m(m+1)}^{p},$$where $I^{p}$ and $L^{p}$ are the interface and layer matrix, respectively. Hence, there are
(4)$$\begin{array}{c} I_{(m-1)m}^{p}=\frac{1}{t_{(m-1)m}^{p}}\left(\begin{array}{cc} 1 & r_{(m-1)m}^{p}\\ r_{(m-1)m}^{p} & 1 \end{array}\right), \end{array}$$and
(5)$$L_{m}^{p}=\left(\begin{array}{cc} e^{i\beta_{m}^{p}} & 0\\ 0 & e^{-i\beta_{m}^{p}} \end{array}\right).$$

Here, $r_{(m-1)m}^{p}$ and $t_{(m-1)m}^{p}$ are reflection and transmission coefficients from layer (*m* − 1) to layer (*m*), and $\beta_{m}^{p}$ is phase retardation induced in layer *m* (*m* = 1, 2, 3, 4). We focus on the special case that the incident light is perpendicular to sample surface in the following calculation. Thus, the reflectance and refractive coefficients are
(6)$$\begin{array}{r} r_{(m-1)m}^{p}=\frac{n_{m-1}^{p}-n_{m}^{p}}{n_{m-1}^{p}+n_{m}^{p}},\\ t_{(m-1)m}^{p}=\frac{2n_{m-1}^{p}}{n_{m-1}^{p}+n_{m}^{p}},\\ \beta_{m}^{p}=\frac{2\pi d_{m}n_{m}^{p}}{\lambda }, \end{array}$$where $n_{m}^{p}$ and $d_{m}^{p}$ are the complex refraction indices and the thickness of layer m, respectively. λ corresponds to incident wavelength. When calculating the optical reflectance along each axis of BP, the refractive indices along the corresponding crystalline axis were used. The same refractive indices of isotropic layers were used for both crystalline axes.

The extinction ratio (ER) was used to evaluate the polarization degree of the polarizer, which was estimated by [27]
(7)$$ER=10\log_{10}\frac{R_{AC}}{R_{ZZ}}=10\log_{10}\frac{1+N}{1-N}.$$

## 3. Results and Discussion

### 3.1. Three-Phase Configuration

An interesting question is whether one can build a BP polarizer in a simple way that is directly depositing BP film on a bulk substrate, which will greatly simplify the analyses and device fabrications. To this end, we studied BP on quartz (0001) and silicon (001), respectively. These substrates were selected as representative substrates with relatively low (e.g., n = 1.46 at 600 nm) and high (e.g., n = 3.93 at 600 nm) refractive indices. For BP on silicon, the calculated optical anisotropy $∆R/R_{AZ}$ was shown by the solid blue line in Figure 2a. The selected excitation wavelength at 600 nm was for demonstration. The oscillation of $∆R/R_{AZ}$ with thickness changes of BP is due to the modulation of optical interference. The complex refractive indices of BP and silicon were referenced from Schuster et al. [31] and Palik et al. [32], respectively. Figure 2a reveals that the numerical difference of reflectances between ZZ and AC direction varied from ∼0.1 to ∼0.01 matched constructive and destructive interferences, respectively. To understand the origin of this weak optical anisotropy $∆R/R_{AZ}$, we plotted the optical reflectances $R_{ZZ}$ and $R_{AC}$ in Figure 2a shown by dashed lines. The reflectances along both crystalline axes did not approach to zero at destructive interference. Therefore, no extremely optical anisotropy occurs. Figure 2b displays the colorful contour of $∆R/R_{AZ}$ as a function of the thickness of BP and excitation wavelength. It reveals BP-Si has only weak polarization effect in the visible and near-infrared range.

As for BP on ${\mathrm{SiO}}_{2}$ substrate, the optical anisotropy $∆R/R_{AZ}$ at 600 nm was shown as the solid blue line in Figure 2c. The abnormal peaks of $∆R/R_{AZ}$ arise periodically, which are attributed to the optical interference. To understand this, we plotted the reflectances along the two principal axes of BP in dashed lines in Figure 2c. The reflectance along ZZ shows strong oscillations varied from ∼0.05 to ∼0.7, while the reflectance along AC direction shows smaller oscillations, but its oscillation strength is getting weaker when BP gets thicker and thicker. It is owing to BP’s relatively larger extinction coefficients along AC direction (see refractive indices of BP in the Appendix A). Thus the observed peaks of $∆R/R_{AZ}$ were introduced by the nearly zero reflectances along ZZ direction at destructive interference. Figure 2d shows the contour of $∆R/R_{AZ}$ as a function of the thickness of BP and incident wavelength.

The difference of $∆R/R_{AZ}$ between air/BP/Si and air/BP/${\mathrm{SiO}}_{2}$ systems indicates that the refractive indices of substrate play important roles to the overall optical anisotropy. To explore the intrinsic relationship between the refractive indices of each layer and the overall optical anisotropy, we presented an approximation for the three-phase system. Figure 3a displays a three-phase model composed of a BP film on a bulk semiconductor substrate in ambient. Here, we considered the light was normally incident on the surface of BP, and the BP layer bounded by air and the substrate. The light rays reflected from the top and bottom surfaces of BP film are intersected at a point P, leading to superposition and interference. If one incident ray is expressed as $E_{0}e^{i\omega t}$, the successively reflected rays could be expressed by appropriately modifying both the amplitude and phase of the initial wave. Referring to Figure 3a, these are
(8)$$\begin{array}{l} E_{1}=\left(r_{01}E_{0}\right)e^{i\omega t},\\ E_{2}=\left(t_{01}t_{01}^{\prime }r_{12}E_{0}\right)e^{i(\omega t-\delta )},\\ E_{3}=\left(t_{01}t_{01}^{\prime }r_{12}^{3}E_{0}\right)e^{i(\omega t-2\delta )}, \end{array}$$and so on. δ is phase difference between successive reflected beams and given by $\delta =4\pi \tilde{n}d/\lambda$, where $\tilde{n}$ and *d* denote the refractive indices and thickness of BP, respectively. The coefficients $t_{\left(m)(m+1\right)}$ and $r_{\left(m)(m+1\right)}$, m=0,1,2, represent the Fresnel coefficients at the interface between layers *m* and m+1. The light intensity $I_{i}$ of the reflected ray is estimated by $I_{i}=E_{i}^{2}$. Please note that the light intensity of the third emerging beam rapidly diminished as its value is determined by $t_{01}^{2}t_{01}^{\prime 2}r_{12}^{6}$ and these Fresnel coefficients are less than 1. Therefore we considered only the first two emerging beams for this more complex situation of multiple reflections. This simplification usually gives a good approximation for the complex multiple reflections if the reflectance of the film $r_{12}$ is small. One can estimate the optical interference of the three-phase model by two waves, $E_{1}$ and $E_{2}$. It comes to the most classical case of two wave source interference. The minimum light intensity occurs at destructive interference, which is estimated by
(9)$$I_{min}=I_{1}+I_{2}-2\sqrt{I_{1}I_{2}}.$$

Thus, the complete cancellation at destructive interference occurs only when $I_{1}=I_{2}$. Using Stokes relation that $t_{01}t_{01}^{\prime }=1-r_{01}^{2}$, we obtained
(10)$$\begin{array}{l} I_{1}=r_{01}^{2}E_{0}^{2},\\ I_{2}={\left(1-r_{01}^{2}\right)}^{2}r_{12}^{2}E_{0}^{2}. \end{array}$$

When ignoring the items of fourth power of Fresnel coefficients, the light intensity of the two waves sources can be simply estimated according to the interface reflectance as
(11)$$\begin{array}{l} I_{1}\approx R_{01}=r_{01}^{2}={\left(\frac{n_{0}-n_{1}}{n_{0}+n_{1}}\right)}^{2},\\ I_{2}\approx R_{12}=r_{12}^{2}={\left(\frac{n_{1}-n_{2}}{n_{1}+n_{2}}\right)}^{2}, \end{array}$$where the $n_{m}\;\left(m=0,1,2\right)$ denotes the refractive indices of each layer. Thus the refractive index of substrate should be $n_{2}=n_{0}$ or $n_{2}=n_{1}^{2}$ in order to approximate $I_{1}=I_{2}$. For $n_{2}=n_{0}$, it requires that BP flake is suspended in air, which will discuss later. However, for the relatively large refractive index of BP, e.g., $n_{ZZ}=4.08+i0.042$ and $n_{AC}=4.27+i0.25$ at the wavelength of 600 nm [31], a suitable semiconductor substrate whose refractive indices are close to $n_{ZZ}^{2}$ or $n_{AC}^{2}$ did not exist under natural conditions. Perhaps other anisotropic 2D materials with a lower refractive index are a better candidate to build resonant nanoscale polarizer.

Figure 3b,c plot the interfacial reflectances of air-BP, BP-${\mathrm{SiO}}_{2}$ and BP-Si along zigzag and armchair directions, respectively. Both the reflectances $R_{ZZ}$ and $R_{AC}$ from BP-${\mathrm{SiO}}_{2}$ are closer to the reflectances of air-BP compared with the reflectances of BP-Si. The refractive index of ${\mathrm{SiO}}_{2}$ is closer to air compared with Si substrate, which accounts for the oscillation amplitude of the reflectance from air/BP/${\mathrm{SiO}}_{2}$ (Figure 2a is larger than the one of air/BP/Si (Figure 2c). This phenomenon leads to the relatively big $∆R/R_{AZ}$ peaks for BP on ${\mathrm{SiO}}_{2}$ substrate.

Figure 4a shows the calculated optical anisotropy of air/BP/air system as a function of the thickness of BP at a wavelength of 600 nm. The refractive index of air was $n_{0}=1$. As shown in the red and black dashed lines, the destructive interferences along armchair and zigzag axes occur at different thicknesses of BP. This phenomenon is because of the birefringence nature of BP. For example, when the thickness of BP at 73 nm, the 1 st (δ=2π) order destructive interference occurs in ZZ direction and the reflectance is 0.0047. On the contrary, the reflectance along AC direction is not at destructive interference, which is 0.12. Thus one can build a 600 nm polarizer with ER around 13.8 dB using a suspended 73 nm-thick BP. Interestingly, the discrepancy between the two phase differences ($\delta_{ZZ}$ and $\delta_{AC}$) increases as thickness increment of BP. The oscillated amplitude along AC axis rapidly decreases because its large absorption index, which leads to the reflectance along AC direction exhibits small oscillation at a baseline of reflectance of air-BP, as shown by the red dashed line in Figure 4. This increased discrepancy is helpful to improve the reflectance efficiency of the proposed polarizer. For example, when the thickness of BP at 294 nm (δ=8π), the reflectance along ZZ axis is 0.037 and reflectance along AC axis is 0.36, which suggests a 600 nm polarizer with high reflectance and ER around 10 dB. Figure 4b shows the contour map of optical anisotropy $∆R/R_{AZ}$ as a function of the thickness of BP and the incident wavelength. Thus, one can choose the polarization wavelengths by just tuning the thickness of BP. However, the reflectance along ZZ direction cannot vanish entirely at destructive interference, which limits the further improvement of extinction ratio for suspended BP polarizer.

### 3.2. Four-Phase Configuration

As discussed above, to further improve the performance of BP polarizer, an additional resonance cavity that enhances the light-matter interactions is essential. Recently, our group proposed an effective design as BP was placed on a Fabry-Perot cavity made of ${\mathrm{SiO}}_{2}/\mathrm{Si}$ [19]. Here, we systematically explore how the thicknesses of ${\mathrm{SiO}}_{2}$ and BP layer affect the polarization wavelength and extinction ratio.

Figure 5a shows the schematic of four-phase system, air/BP/${\mathrm{SiO}}_{2}$/Si. As demonstration, we firstly set the thickness of ${\mathrm{SiO}}_{2}$ layer at 90 nm and the excitation wavelength at 600 nm. Figure 5b shows the optical anisotropy varies with the thickness of BP. The 1st order destructive interferences along AC and ZZ axes occur both at a thickness of BP around 75 nm, which gives rise to a small $∆R/R_{AZ}$ of 0.89, and is not good for building a polarizer. As the thickness of BP increases, the 2nd order destructive interference along the AC axis occurs at a thickness of BP at 142 nm. On the contrary, the 2nd order destructive interference along the ZZ axis occurs at a thickness of BP at 148 nm. A BP polarizer with ER of 12.6 dB can be obtained using 148 nm BP on 90 nm ${\mathrm{SiO}}_{2}/\mathrm{Si}$. When the thickness of BP is 222 nm, 3 rd order destructive interference along ZZ axis, $∆R/R_{AZ}$ reaches to 1.988, and the ER arrives 25.2 dB. At the same time, the reflectances along AC axis are ∼30%. As the 4th order destructive interference along ZZ occurs, the ER of BP polarizer reaches 21.8 dB, and the reflectance along AC axis is ∼35%.

Figure 5c shows the colorful contour of optical anisotropy $∆R/R_{AZ}$ of BP on 90 nm ${\mathrm{SiO}}_{2}/\mathrm{Si}$ substrate as a function of BP thickness and incident wavelength. The extraordinary $∆R/R_{AZ}$ arises when the destructive interferences occur in ZZ direction. Figure 5d plots the peak wavelengths of $∆R/R_{AZ}$ that ER are higher than 16 dB as a function of BP thickness. One can build a nanoscale polarizer (ER>16 dB) at a broad and continuous wavelength range from 477 nm to 900 nm.

We also calculated the optical anisotropy $∆R/R_{AZ}$ of BP on 300 nm ${\mathrm{SiO}}_{2}/\mathrm{Si}$ substrate. Figure 5e plots $∆R/R_{AZ}$ as a function of BP thickness and incident wavelength. Figure 5f plots the peak wavelengths of $∆R/R_{AZ}$ that ER are higher than 16 dB varies with the thickness of BP. Thus, one can build a polarizer only at a wavelength range from 509 nm to 710 nm when a 300 nm ${\mathrm{SiO}}_{2}/\mathrm{Si}$ substrate was applied.

To explore the relationship between the polarization wavelength and the thickness of ${\mathrm{SiO}}_{2}$ layer, we calculated the optical anisotropy $∆R/R_{AZ}$ as a function of incident wavelength and ${\mathrm{SiO}}_{2}$ thickness. We set the thickness of BP at 125 nm, 150 nm, and 175 nm, respectively. As shown in Figure 6a,b, if the thickness of BP is 125 nm, the polarized wavelengths continuously and periodically vary at a specific range of 530 nm–549 nm and 860 nm–880 nm when regularly changing the ${\mathrm{SiO}}_{2}$ thickness. Similarly, the polarized wavelengths change periodically at a particular range of 589 nm–633 nm and 662 nm–719 nm if BP is 150 nm and 175 nm, respectively. Compared with calculation results in Figure 5, the thickness of BP is the primary determinant for the polarization wavelengths of BP-${\mathrm{SiO}}_{2}$-Si configuration.

Considered the non-linear dependence of the polarization wavelength on the thicknesses of BP and ${\mathrm{SiO}}_{2}$ layers, we iterated the $∆R/R_{AZ}$ with BP thickness varied from 10 nm to 250 nm and ${\mathrm{SiO}}_{2}$ thickness varied from 0 to 500 nm. Figure 7a,b shows the contour map of polarization wavelengths of $∆R/R_{AZ}$ with ER higher than 10 dB and 16 dB, respectively. This calculation confirms the feasibility of constructing a polarizer (ER > 16 dB) in visible and near-infrared regime by just tuning the thicknesses of BP and ${\mathrm{SiO}}_{2}$ layers. This plot is a guide for the nanoscale polarizer. Interestingly, the polarization wavelength shows independence on the thickness of ${\mathrm{SiO}}_{2}$ when setting the ${\mathrm{SiO}}_{2}$ thickness of 50 nm. Considering the hard control of BP thickness, the mild dependence of polarization wavelength on BP thickness is a benefit for designing the polarizer at a given wavelength. Therefore this makes ${\mathrm{SiO}}_{2}$ thickness of approximately 50 nm most appropriate to be used as the substrate.

### 3.3. Five-Phase Configuration

It is well known that the fast oxidization and degradation in ambient are the major hurdle for BP [33,34,35]. To deal this obstacle, sandwich structure, including encapsulations by atomic-thick ${\mathrm{Al}}_{2}O_{3}$ [33,34] or *h*-BN [35], and BP-based heterostructures, were proved to be one of the most simple and successful methods. Besides doping and chemical functionalization have been widely used to passivate BP against degradation in ambient conditions [36,37,38]. Among them, the insulating *h*-BN can be an ideal substrate and protect layer because of its atomically smooth surface yet no dangling bond, chemical inertness and high temperature sustainability [39,40]. Here, we calculated optical anisotropy of a five-phase system of air/*h*-BN/BP/${\mathrm{SiO}}_{2}$/Si, where the atomic-thick *h*-BN is the protective layer [35], as shown in Figure 8a. We set two thicknesses of *h*-BN. One case is at 6 nm which introduces small interference. Alternatively, a 30 nm *h*-BN was used as an example of a relatively thick film which induces large interference. For each case, we extracted the total optical anisotropy $∆R/R_{AZ}$ as BP thickness varying from 10 to 250 nm and ${\mathrm{SiO}}_{2}$ thickness varying from 0 to 500 nm. Figure 8b,c show the contours of the wavelengths of $∆R/R_{AZ}$ with ER higher than 16 dB when the thicknesses of *h*-BN were 6 nm and 30 nm, respectively. Figure 8b indicates the 6 nm-*h*-BN did not have severe impact on $∆R/R_{AZ}$ compared with bare BP on Si-${\mathrm{SiO}}_{2}$ system in Figure 7b. However, Figure 8c illustrates the 30 nm-*h*-BN has a strong effect on the $∆R/R_{AZ}$, especially for the thickness region of BP at 1st order destructive interference (δ=2π). Besides, the additional 30 nm-*h*-BN layer weakens the dependence of polarization wavelength on the thickness of ${\mathrm{SiO}}_{2}$ layer, which opens up a new route to regulate the polarization wavelength using more complex multi-layer heterostructures.

## 4. Conclusions

Using the scattering matrices method, we computed the optical anisotropy of BP/Si, BP/${\mathrm{SiO}}_{2}$, BP/${\mathrm{SiO}}_{2}$/Si, and *h*-BN/BP/${\mathrm{SiO}}_{2}$/Si in the near-infrared and visible range. The calculation reveals that an additional resonance cavity is essential to achieve high extinction ratio. The BP-based polarizer, with polarization wavelength continuously changes from 400 nm to 900 m and ER > 16 dB, can be achieved through both BP/${\mathrm{SiO}}_{2}$/Si and *h*-BN/${\mathrm{SiO}}_{2}$/Si configurations. The ∼50 nm ${\mathrm{SiO}}_{2}/\mathrm{Si}$ is the most suitable substrate because of the mildest dependence of polarization wavelength on layer thickness, which benefits for fabrication considering the difficulties of precise thickness control of each layer. Furthermore, a layer of h-BN on top of the BP/${\mathrm{SiO}}_{2}$/Si structure even increases the polarization performance, which promises an effective protection to the degradation of BP from oxidation and strengths the practical applications of BP polarizer. Our methods are also compatible with other anisotropic 2D materials, which establishes the path for developing nanoscale polarizer through cavity-based 2D heterostructures.

The vertical heterostructure composed of one or more anisotropic 2D materials provides potential solutions to design diverse high-performance nano polarizer, possessing the advantage of design freedom and more straightforward fabrication. Multi-heterostructure weakens the standalone influence of each layer, which implies a route for broadband polarizer by matching the thickness and refractive indices of each layer. The accurate evaluation of the refractive indices of each layer is crucial for the design, which affects the result. However, the effective and accurate determination of the refractive indices of the rising anisotropic 2D is still lack of investigation.

## Figures and Tables

**Figure 1 nanomaterials-09-00168-f001:**
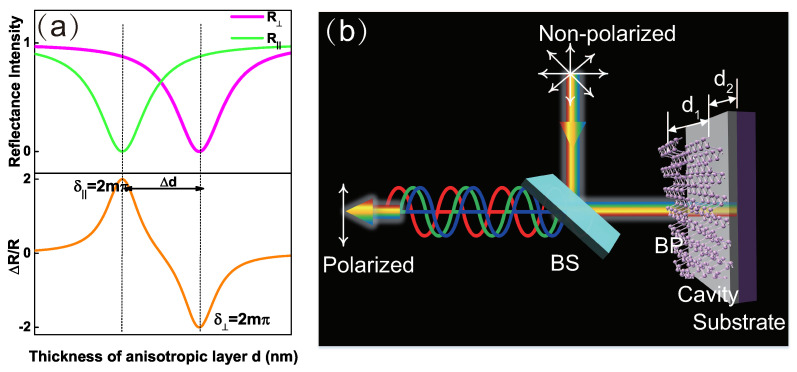
(**a**) Principle illustration. The upper panel shows that the reflections along two mutually orthogonal crystal axes asynchronously vary with the thickness of the birefringence layer. The lower panel plots the corresponding $∆R/R=2\left(R_{\left|\right|}-R_{⊥}\right)/\left(R_{\left|\right|}+R_{⊥}\right)$, which shows that destructive interference contributes to the extreme values of ∆R/R. (**b**) Scheme of the proposed reflective polarizer based on BP film and a Fabry-Perot cavity.

**Figure 2 nanomaterials-09-00168-f002:**
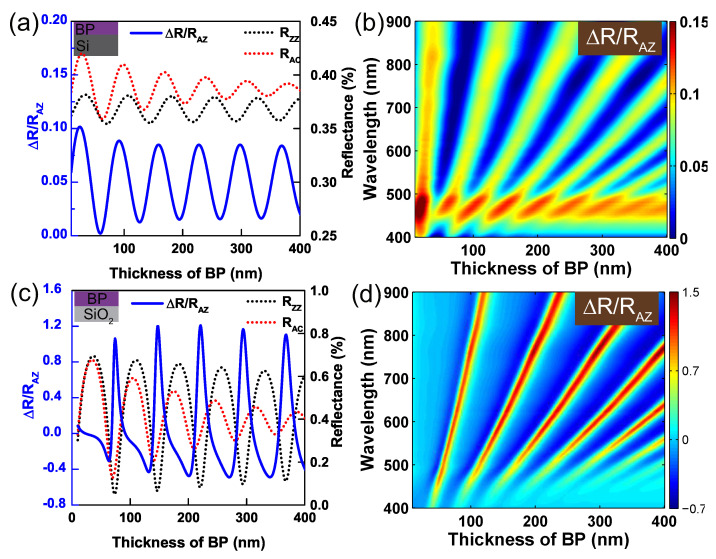
The optical anisotropy of BP on Si (**a**) and ${\mathrm{SiO}}_{2}$ (**c**) substrate as a function of the thickness of BP film at a wavelength of 600 nm. The blue solid lines denote the optical anisotropy $∆R/R_{AZ}$, and the black and red dashed lines represent the reflectances along ZZ and AC directions, respectively. Contour plot of optical anisotropy $∆R/R_{AZ}$ of BP on Si (**b**) and ${\mathrm{SiO}}_{2}$ (**d**) substrate varied as a function of the thickness of BP film and the wavelength of incident light.

**Figure 3 nanomaterials-09-00168-f003:**
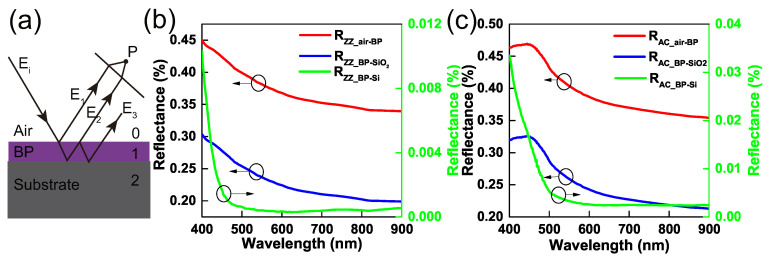
(**a**) Schematic diagram for the reflected beams in the three-phase thin film system: air, BP (cavity layer) and opaque substrate (bottom layer). When the BP flake is thick enough, it forms as a resonant cavity and the optical interference occurs from the upper and lower surfaces of BP film. The incident light is perpendicular to the sample plane and the oblique drawing of the light beam is for the convenience of illustration. (**b**,**c**) Comparison of optical reflectances of BP-Si and BP-${\mathrm{SiO}}_{2}$ interface. The reflectances are plotted along zigzag and armchair directions, respectively.

**Figure 4 nanomaterials-09-00168-f004:**
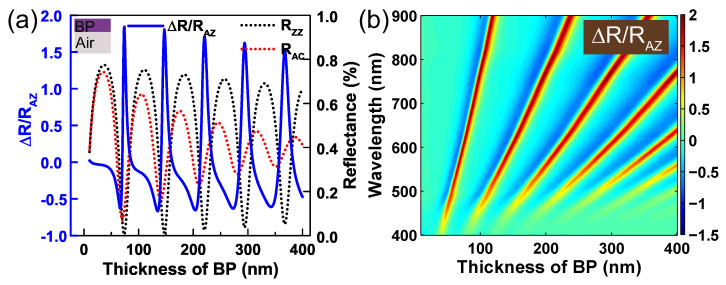
(**a**) The optical anisotropy of suspended BP as a function of the thickness of BP at a wavelength of 600 nm. The blue solid line denotes the optical anisotropy $∆R/R_{AZ}$, and the black and red dashed lines represent the reflectances along zigzag and armchair directions, respectively. (**b**) Contour of optical anisotropy $∆R/R_{AZ}$ of suspended BP as a function of the thickness of BP film and the wavelength of incident light.

**Figure 5 nanomaterials-09-00168-f005:**
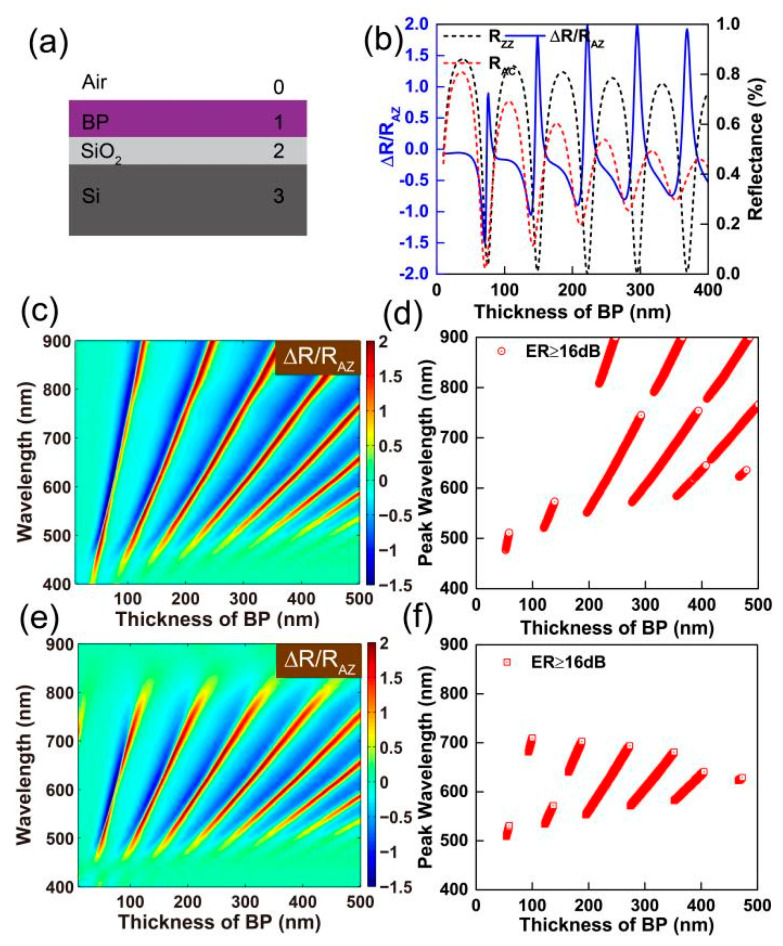
(**a**) A schematic diagram for a four-phase thin film system: air, a dielectric film (BP and SiO_2_), and a Si substrate. (**b**) The optical anisotropy of BP on a 90 nm SiO_2_/Si substrate as a function of the thickness of the BP film at a wavelength of 600 nm. Left: contour plot of ∆*R*/*R_AZ_* of BP on a 90 nm (**c**) and 300 nm (**e**) SiO_2_/Si substrate as a function of the thickness of BP and the wavelength of incident light. Right: plots of polarization wavelengths with an extinction ratio higher than 16 dB as a function of the thickness of the BP film on a 90 nm (**d**) and 300 nm (**f**) SiO_2_/Si substrate.

**Figure 6 nanomaterials-09-00168-f006:**
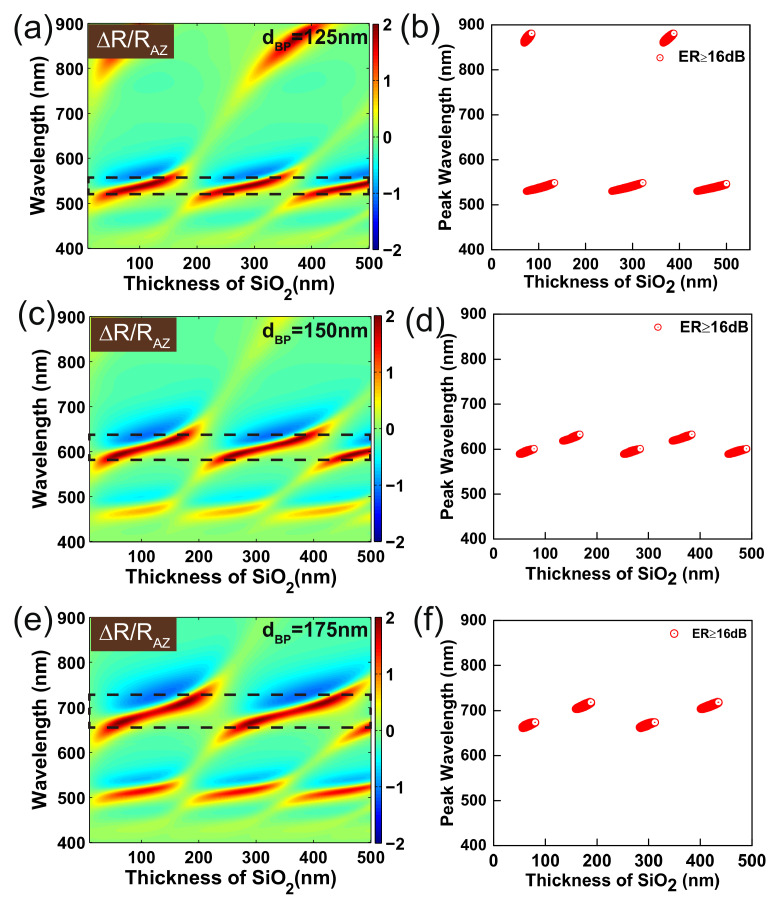
Right column: contour of $∆R/R_{AZ}$ varies with the wavelength of incident light and the thickness of ${\mathrm{SiO}}_{2}$. The thicknesses of BP are (**a**) 125 nm, (**c**) 150 nm and (**e**) 175 nm, respectively. Left column: plots of peak wavelengths with extinction ratio higher than 16 dB as a function of the thickness of ${\mathrm{SiO}}_{2}$ with the thickness of BP at (**b**) 125 nm, (**d**)150 nm and (**f**) 175 nm.

**Figure 7 nanomaterials-09-00168-f007:**
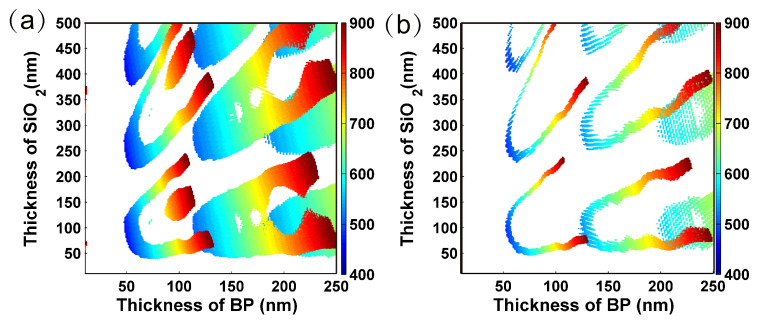
Contour map of the polarized wavelength of $∆R/R_{AZ}$ that extinction ratio are higher than (**a**) 10 dB and (**b**) 16 dB as a function of the thickness of BP and ${\mathrm{SiO}}_{2}$ layer.

**Figure 8 nanomaterials-09-00168-f008:**
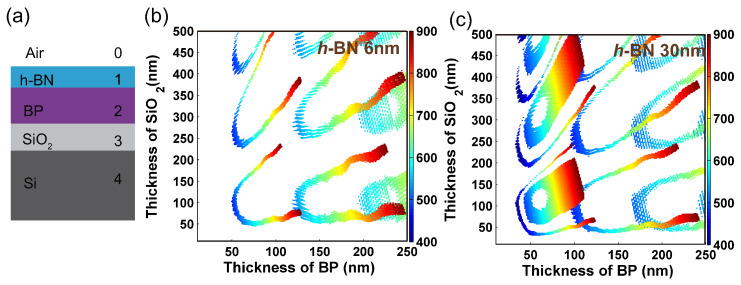
(**a**) Schematic diagram of five-layer heterostructure. Contour maps of the wavelength of $∆R/R_{AZ}$ with extinction ratio higher than 16 dB as a function of the thicknesses of ${\mathrm{SiO}}_{2}$ and BP film: (**b**) 6 nm and (**c**) 30 nm *h*-BN layer are placed on the top of BP layer, respectively.

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
