# Peer review of "Black Phosphorus Nano-Polarizer with High Extinction Ratio in Visible and Near-Infrared Regime"

_nanomaterials, 2019, doi:10.3390/nano9020168_

Reviewer 1 Report

The authors have computationally examined the design of a high extinction ratio nano polarizer based on 1 black phosphorus (BP). A scattering-matrix calculation method is applied to compute the overall polarization extinction ratio along two orthogonal directions. The results reveal that, with a resonance

cavity of SiO2, both BP-SiO2-Si and h-BN-BP-SiO2-Si configurations can build a linear polarizer with extinction ratio higher than 16dB at a polarized wavelength in the range of 400nm-900nm. This is an interesting study and is presented well and I recommend publication with minor revisions.

Can the authors comment on the choice of hBN as the protective layer. Can they provide a short discussion on this and the differences between the other stabilisation techniques. They can refer to papers such as Advanced Materials 29 (27), 1700152 for a short discussion.

It is also known that the UV blue spectrum of the light causes the maximum amount of material damage (npj 2D Materials and Applications 1 (1), 18). Does h-BN protect against this photo oxidation.

Does the polarisation effect is BP get affected by the formation of oxide species which typically form on the surface straight after synthesis. Can the authors provide a short comment.

Author Response

Comments: The authors have computationally examined the design of a high extinction ratio nano polarizer based on black phosphorus (BP). A scattering-matrix calculation method is applied to compute the overall polarization extinction ratio along two orthogonal directions. The results reveal that, with a resonance cavity of SiO2, both BP-SiO2-Si and h-BN-BP-SiO2-Si configurations can build a linear polarizer with extinction ratio higher than 16dB at a polarized wavelength in the range of 400nm-900nm. This is an interesting study and is presented well and I recommend publication with minor revisions.

Point 1: Can the authors comment on the choice of hBN as the protective layer. Can they provide a short discussion on this and the differences between the other stabilisation techniques? They can refer to papers such as Advanced Materials 29 (27), 1700152 for a short discussion.

Response 1: Thank you for your suggestion. We add a sentence of “Besides, doping and chemical functionalization have been widely utilized to passivate BP against degradation in ambient conditions [36-38]. Among them, the insulating h-BN can be an ideal substrate and protect layer because of its atomically smooth surface yet no dangling bond, chemical inertness, and high temperature sustainability [39, 40].” in the beginning paragraph of section 3.3.

Point 2: It is also known that the UV blue spectrum of the light causes the maximum amount of material damage (npj 2D Materials and Applications 1 (1), 18). Does h-BN protect against this photo oxidation.

Response 2: Thank you for your comment. It is a very interesting question as the degradation mechanism is still in debate. The report by Ahmed T et.al. (Ahmed T, Balendhran S, Karim M N, et al. Degradation of black phosphorus is contingent on UV–blue light exposure[J]. npj 2D Materials and Applications, 2017, 1(1): 18.) points out a very important conclusion that the UV and blue light causes severe damage for the materials compared with other component of spectrum with longer wavelength. Considering the transparent property of h-BN, the UV and blue lights will penetrate through the covered h-BN layer. Fortunately, both experimental and theoretical analyses demonstrate that the oxygen and the light are both essential for the degradation of BP and the water accelerates the oxidization of BP after the first chemical reaction between BP and oxygen. ([1]Zhou Q, Chen Q, Tong Y, et al. Angewandte Chemie International Edition, 2016, 55(38): 11437-11441.[2] Walia S, Sabri Y, Ahmed T, et al. 2D Materials, 2016, 4(1): 015025.) Thus, the capped layer, such as h-BN, isolates BP from the air, which might be also used to protect BP against the photo oxidation of UV and blue light. However, this kind of experiment has not been conducted to the best of our knowledge, which is an interesting issue. However, the encapsulation method has been proven to be an effective approach to protect BP from degradation. (e.g. [1] Wood J D, Wells S A, Jariwala D, et al. Nano Letters, 2014, 14(12): 6964-6970. [2] Doganov R A, O’Farrell E C T, Koenig S P, et al. Nature communications, 2015, 6: 6647. [3] Illarionov Y Y, Waltl M, Rzepa G, et al. ACS nano, 2016, 10(10): 9543-9549;)

Point 3: Does the polarisation effect is BP get affected by the formation of oxide species which typically form on the surface straight after synthesis. Can the authors provide a short comment?

Response 3: Thank you for your comment. We believe the slight oxidization on the top of BP will affect the polarization effect. According to recent studies, the oxidization layer of BP did not show anisotropic properties, which is easy understood because the main components of oxide species are mostly amorphous, like P2O5 or P4O10. Thus, the polarization effect caused by oxide species can be estimated through the convenient scattering-matrix calculation method, in which the oxidization layer was modeled through the effective medium approximation (EMA) method. The isotropic oxide species have similar polarizer effect with other isotropic hybrid layer, such as h-BN. However, more detailed experiments are needed to confirm the isotropic properties of oxide species.

Reviewer 2 Report

Dear Editor,

The present manuscript faces, rigurously and clearly, the eventual role that an anisotropic material as black-phosphorus may play as a nano-polarizer. Such a role is one of the most evident roles that anisotropic semiconductors may play in optoelectronics. However, this eventual application had not been surprisingly analysed since now. For this reason, I consider the subject of this manuscript very suitable for Nanomaterials. This fact, together with the high quality of results obtained as well as their clear presentation and discussion, makes me to suggest the present manuscript for publication in its present form.

Only few minor remarks to authors.

When authors present Fig. 3a (line 109), they show a non-normal incidence situation. However, they inmediately, after presentation of Fig. 3a, they mention that they limite themselves to normal incidence of light. It is not clear for me if they have studied the case of non-normal incidence. Please, clarify if all results reported are limited to normal incidence of light.

The fact that black-P must be combined with a appropiately designed substrate to enhance the role of of black-P as nanopolarizers seems to suggest that anisotropic 2D materials, alone, hardly achieve to act as a nanopolarizer. Could authors introduce some discussion about  conditions that other 2D materials may satisfy to present stronger properties as nanopolarizers than black-P?

Author Response

Comments: The present manuscript faces, rigurously and clearly, the eventual role that an anisotropic material as black-phosphorus may play as a nano-polarizer. Such a role is one of the most evident roles that anisotropic semiconductors may play in optoelectronics. However, this eventual application had not been surprisingly analysed since now. For this reason, I consider the subject of this manuscript very suitable for Nanomaterials. This fact, together with the high quality of results obtained as well as their clear presentation and discussion, makes me to suggest the present manuscript for publication in its present form. Only few minor remarks to authors.

Point 1: When authors present Fig. 3a (line 109), they show a non-normal incidence situation. However, they inmediately, after presentation of Fig. 3a, they mention that they limite themselves to normal incidence of light. It is not clear for me if they have studied the case of non-normal incidence. Please, clarify if all results reported are limited to normal incidence of light.

Response 1Thank you for your suggestion. We are sorry for this mistake. We add the sentence of “We focus on the special case that the incident light is perpendicular to sample surface in the following calculation.” in front of Eq.(6). Besides, we add the sentence of “The incident light is perpendicular to the sample plane and the oblique drawing of the light beam is for the convenience of illustration.” in the caption of Fig. 3(a).

Point 2: The fact that black-P must be combined with a appropriately designed substrate to enhance the role of black-P as nanopolarizers seems to suggest that anisotropic 2D materials, alone, hardly achieve to act as a nanopolarizer. Could authors introduce some discussion about conditions that other 2D materials may satisfy to present stronger properties as nanopolarizers than black-P?

Response 2:  The reviewer is right. It is hard to build high-performance nano-polarizer through BP alone. This is mainly because of the mismatch between BP and substrate. Especially, a suitable semiconductor substrate did not exist under natural condition because of the large refractive index of BP. This part was discussed in section 3.1 in the manuscript and the deduced condition for high-performance nanopolarizer was presented in the paragraph after Eq.11.

 Besides, two more general rules can be considered when selecting anisotropic 2D materials to build nanopolarizer. Firstly, the numerical difference between the refractive indices of two crystalline axes of anisotropic 2D materials should be larger enough. Secondly, the reflectance at destructive interference determines the extinction ratio and the less reflectance at destructive interference contributes to higher extinction ratio.  Considering the last two more rules are general for all polarizers, we prefer that we did not emphasize these in the manuscript.

Reviewer 3 Report

Manuscript # Nanomaterials-424533: “Black Phosphorous Nano-Polarizer with High Extinction Ratio in Visible and Near-infrared Regime

W. Shen and co-authors have investigated the optical anisotropy and polarizer performance of various BP-based multilayer nano-polarizers including BP-Si, BP-SiO2, BP-SiO2-Si, and h-BN-BP-SiO2-Si by the scattering matrices method. Their calculations reveal that high extinction ratio (ER) can be obtained by plugging resonance cavity into the structures. More interestingly, the most practical configuration of h-BN-BP-SiO2-Si where the BP layer is prevented from being damaged in the ambient atmosphere by coating BP with h-BN shows better performance in comparison of those without h-BN. Considering that the calculations and analyses have been performed systematically and rigorously and the results are interesting and useful for the community, some minor concerns should be addressed for the publication in Nanomaterials.

My detailed concerns are addressed as follows:

1.      There are some grammatical errors throughout the manuscript. It should be proof-read by a native English speaker.

2.      Considering that the calculations have been performed with the light normally incident on the surface of the BP layer, the situation is described wrongly in the Fig. 3 (a). 

3.      Is there any particular reason why only the two thicknesses of h-BN, 6 nm and 30 nm, were chosen for the calculations?

4.      In the Conclusions section, h-BN-SiO2-Si should be corrected as h-BN-BP-SiO2-Si.

5.      I would like to suggest authors to use “/” instead of “-” for the symbols separating layers since the word h-BN also includes “-”.

Author Response

Comments: W. Shen and co-authors have investigated the optical anisotropy and polarizer performance of various BP-based multilayer nano-polarizers including BP-Si, BP-SiO2, BP-SiO2-Si, and h-BN-BP-SiO2-Si by the scattering matrices method. Their calculations reveal that high extinction ratio (ER) can be obtained by plugging resonance cavity into the structures. More interestingly, the most practical configuration of h-BN-BP-SiO2-Si where the BP layer is prevented from being damaged in the ambient atmosphere by coating BP with h-BN shows better performance in comparison of those without h-BN. Considering that the calculations and analyses have been performed systematically and rigorously and the results are interesting and useful for the community, some minor concerns should be addressed for the publication in Nanomaterials.

My detailed concerns are addressed as follows:

Point 1: There are some grammatical errors throughout the manuscript. It should be proof-read by a native English speaker.

Response 1: We are sorry for this. We have checked the grammar for the whole manuscript and the changes were marked by red color.

 Point 2: Considering that the calculations have been performed with the light normally incident on the surface of the BP layer, the situation is described wrongly in the Fig. 3 (a).  

Response 2: We are sorry for this mistake. To clarify this, we add the sentence of “The incident light is perpendicular to the sample plane and the oblique drawing of the light beam is for the convenience of illustration.” in the caption of Fig. 3(a). In order to avoid further misunderstanding to readers, we add the sentence of “We focus on the special case that the incident light is perpendicular to sample surface in the following calculation.” in front of Eq.(6).

Point 3: Is there any particular reason why only the two thicknesses of h-BN, 6 nm and 30 nm, were chosen for the calculations? 

Response 3: Thank you for your comment. We selected 6nm and 30nm h-BN as two typical demonstrations of relatively thin and thick thickness.

Point 4: In the Conclusions section, h-BN-SiO2-Si should be corrected as h-BN-BP-SiO2-Si.

Response 4: We are sorry for this mistake. We have changed it.

 Point 5: I would like to suggest authors to use “/” instead of “-” for the symbols separating layers since the word h-BN also includes “-”.

Response 5: Thank you for your suggestion. We have replaced “-” with “/” for separating layers for the whole manuscript.
